# The Influence of Whey Protein on Muscle Strength, Glycemic Control and Functional Tasks in Older Adults with Type 2 Diabetes Mellitus in a Resistance Exercise Program: Randomized and Triple Blind Clinical Trial

**DOI:** 10.3390/ijerph20105891

**Published:** 2023-05-20

**Authors:** André Luiz de Seixas Soares, Adriana Machado-Lima, Guilherme Carlos Brech, Júlia Maria D’Andréa Greve, Joselma Rodrigues dos Santos, Thiago Resende Inojossa, Marcelo Macedo Rogero, João Eduardo Nunes Salles, José Maria Santarem-Sobrinho, Catherine L. Davis, Angelica Castilho Alonso

**Affiliations:** 1Program in Aging Sciences, Universidade São Judas Tadeu (USJT), São Paulo 03166-000, SP, Brazil; 2Georgia Prevention Institute, Medical College of Georgia, Augusta University, Augusta, GA 30912, USA; 3Laboratory Study of Movement, Instituto de Ortopedia e Traumatologia do Hospital das Clínicas (IOT-HC), Faculdade de Medicina da Universidade de São Paulo (FMUSP), São Paulo 05402-000, SP, Brazil; 4Department of Nutrition, Faculty of Public Health, FMSUP, São Paulo 01151-000, SP, Brazil; 5Department of Internal Medicine, The Discipline of Endocrinology, Santa Casa of São Paulo Medical School, São Paulo 01224-001, SP, Brazil; 6Centro de Estudos em Ciências da Atividade Física-CECAFI-FMUSP, São Paulo 05403-010, SP, Brazil

**Keywords:** type 2 diabetes, resistance training, postural balance, older adults, whey protein

## Abstract

Objectives: To evaluate the effect of whey protein (WP) supplementation associated with resistance training (RT) on glycemic control, functional tasks, muscle strength, and body composition in older adults living with type 2 diabetes mellitus (T2DM). Secondly, to evaluate the safety of the protocol for renal function. Methods: The population comprised twenty-six older men living with T2DM (68.5 ± 11.5 years old). The participants were randomly assigned to the Protein Group (PG) and the Control Group (CG). The handgrip test and evolution of exercise loads, according to the Omni Resistance Exercise Scale, evaluated muscle strength. Functional tasks were assessed by force platform in three different protocols: Sit-to-Stand, Step/Quick Turn, and Step Up/Over. Body composition was evaluated by bioimpedance and glycemic control and renal function were assessed by biochemical analyses. Both groups performed RT for 12 weeks, twice a week, prioritizing large muscle groups. Protein supplementation was 20 g of whey protein isolate and the CG was supplemented with an isocaloric drink, containing 20 g of maltodextrin. Results: There was a significant difference in muscle strength, according to the evolution of the exercise loads, but it was not confirmed in the handgrip test. However, there was no significant difference between the groups, regarding performance in functional tasks, glycemic control, or body composition. Renal function showed no alteration. Conclusion: The intake of 20 g of WP in older male adults living with T2DM did not increase the effect of RT on muscle strength, functional tasks, and glycemic control. The intervention was proven safe regarding renal function.

## 1. Introduction

Diabetes mellitus (DM) is a chronic disease characterized by metabolic disorders, mainly due to inadequate regulation of blood glucose. This is associated with damage to the mechanisms of production and/or use of the hormone insulin, which is primarily responsible for transporting glucose from the bloodstream to cells [1]. Type 2 diabetes mellitus (T2DM), which represents around 90% of all cases of diabetes [2], has its most significant risk factors in socio-demographic changes, such as increased life expectancy, physical inactivity, ultra-processed foods, obesity, and urbanization [3]. In 2021, there were five hundred thirty-seven million adults living with DM worldwide, and projections are for seven hundred eighty-three million or 12.2% of the world’s population in 2045. Among older adults, estimates for 2045 reach the number of two hundred seventy-six million six hundred thousand individuals living with the disease [4].

There is a natural increase in muscle protein hydrolysis, associated with a decrease in its synthesis during aging. This is a description of anabolic resistance, potentially aggravated in people living with T2DM, due to the worsening of the insulin and intracellular anabolic signaling pathways [1,5,6].

Resistance training (RT) can positively intervene during the aging process and in several distinct mechanisms of T2DM pathophysiology, emphasizing the improvement in sarcopenia and/or dynapenia and glycemic control. In addition, RT can be an essential therapeutic ally to minimize the chances of serious outcomes, involving loss of autonomy, falls, decreased quality of life, and metabolic disorders that can threaten multiple systems [7,8,9].

There is an important body of evidence that shows supplementation with WP can optimize the anabolic effect of RT in older adults, thus promoting an increase in lean mass with a decrease in total body mass [5,10]. These changes can already produce positive results on glycemia through multiple mechanisms, including mitochondrial biogenesis, an increase of glucose transporter 4 expression (GLUT4), and augmented glycogen synthesis, among several others [11]. Moreover, WP supplementation may directly improve glycemic control by optimizing the pathway of incretins, such as gastric inhibitory polypeptide (GIP) and glucagon-like peptide-1 (GLP-1) [12]. WP can also inhibit alpha-glucosidase, which mediates the intestinal absorption of glucose, and dipeptidyl peptidase-IV (DPP-4), an enzyme that degrades incretins [13,14]. In addition, there are anorexigenic effects due to vagal stimulation in a direct influence on the hypothalamus, a reduction in ghrelin and its orexigenic effect, and an increase in satiety by modulating the speed of gastric emptying, and also by increasing GLP-1 levels [14,15].

Recently, a systematic review comprising eleven studies was published combining diet, supplements, and exercises [16], and in only two of the studies, protein supplementation [17,18] in people living with DMT2 did not respond to a certain type of nutritional stimulation. However, on a positive note, recent data have shown that increasing the dose of the supplement or extending the intervention period and the type of exercise can lead to better clinical outcomes, which should be investigated further. Therefore, more studies are needed to better understand how to optimally combine these three factors for the management of T2DM.

The positive effects of RT, WP supplementation, and the combination of both in glycemic control, body composition, metabolic profile, inflammation, strength, and potential balance are well documented [16,17,18,19,20]. Accordingly, the importance of this study resides in the contribution that it can offer a better understanding of doses, frequency, volume, and intensity of these resources to provide more efficient and safe strategies that can benefit the globally increasing population of older adults living with T2DM.

Therefore, the hypothesis is that the effects of this supplementation are synergistic with RT and optimize a set of desirable adaptations in older adults living with T2DM. Hence, the present study aims to evaluate the effect of protein supplementation associated with RT for 12 weeks on glycemic control, functional tasks, muscle strength, and body composition in older adult men living with T2DM and secondarily, to evaluate the safety of the protocol, concerning renal function.

## 2. Materials and Methods

Study design and population: This is a randomized, triple-blinded clinical trial (Registered in ClinicalTrials.gov, identifier: NCT03792646, developed at Universidade São Judas Tadeu in partnership with the Laboratory of the Study of Movement of the Institute of Orthopedics and Traumatology (IOT) of Hospital das Clínicas (HC) of the Faculty of Medicine of the University of São Paulo (FMUSP). Approved by the Ethics Committee of the FMUSP n° CAAE: 39202214.8.0000.0065.

Participants: The population of the present study consisted of 28 men with T2DM, aged 68.2 (±4.0) years. Participants were recruited from senior groups at the Laboratory for the Study of Movement (LEM) with the first contact made by telephone call. After verifying potential eligibility, they were invited to come to LEM for the interview and first set of assessments.

For inclusion, the study had the following criteria: living with T2DM on a stable dose of medication (oral antidiabetic agents or insulin) for over a year, availability to participate in the RT program for three months at the University Hospital, Glycated hemoglobin (HbA1c) between 6.0 and 9.0%, and renal function assessed by the Modification of Diet in Renal Disease (MDRD) method above 60 mL/h. Similarly, liver function was assessed by the blood levels of Aspartate aminotransferase (AST) and Alanine aminotransferase (ALT), which was expected to be up to 2.5 times higher than the limit of the method, and no impairment of the musculoskeletal system, such as pain or any condition that could cause inability to sustain a whole training session without restrictions, according to the protocol. They also could not have any decompensated or untreated chronic disease. As exclusion criteria, the study established the inability to carry out assessments, and during the training period, the participants could not have more than three absences or any related adverse event, such as an outcome of musculoskeletal pain or worsening of any prior disease symptoms or signs. 

After this approach, all the participants signed the Informed Consent (IC) form and were invited for a second set of evaluations.

The selected participants had to undergo an evaluation by a physician to check clinical conditions and safety to engage in RT.

Laboratory analysis: Blood samples were also requested for laboratory analyses from the participants who were clinically approved. Peripheral blood samples (20 mL) were collected in tubes, containing EDTA anticoagulant at the baseline and at the end of 12 weeks of protocol. Whole blood with EDTA was centrifuged to obtain plasma, which was used to evaluate the circulating concentration of glucose, insulin, fructosamine, hepatic enzymes, and glycated hemoglobin (HbA1c), using commercial kits, according to the manufacturer’s instructions (Labtest, Minas Gerais, Brazil).

Renal function was obtained by the Modification of Diet in Renal Disease (MDRD) method, using the formula: glomerular filtration rate GFR = 175 × Serum (Cr) (−1.154) × age (−0.203) × 1.212 (if black) × 0.742 (if female). Moreover, insulin resistance was calculated by the HOMA-IR formula: (Homa-IR: [(Glucose (mg/dL) × (0.0555) × Insulin (UI/mL)] ÷ 22.5) [16,17].

Clinical assessment: Socio-demographic characterization was performed using a questionnaire, regarding age, education, profession, marital status, physical activity, and current health conditions.

Assessment of muscle strength (1RM): The 1RM test was defined as the maximum amount of weight lifted in a simple maximum effort, in which each participant completed an entire movement that could not be repeated a second time. The maximum load was the last one, in which each participant performed a movement with adequate performance standards. All of the participants in the study participated in two sessions of the 1RM test. The first one was for the participants to become familiar with the test, and the second one was for consideration of valid results.

Evaluation of Handgrip Strength (HGS): For this evaluation, a Jamar^®^ manual dynamometer was used. For the procedure, each participant remained seated in a chair without armrests, with their feet flat on the floor, and their hips and knees flexed at 90°, with shoulders positioned adducted and in the neutral position for rotation, and elbows in 90° of flexion with forearms and wrists in the neutral position. Hands were alternated for each maneuver, leaving one minute of rest between tests, with each first maneuver performed with the dominant limbs. The outcome of this analysis was given in kg/force by the average of the three attempts of each evaluated limb [18].

Assessment of functional tasks: To assess dynamic balance NeuroCom Balance Master^®^ equipment (NeuroCom International, Inc., Clackamas, OR, USA) was used, including a computer with a force platform to record information through piezoelectricity transducers. The force platform information included X (±0.08 cm) and Y (±0.25 cm) positions of the vertical force center and the total vertical force (±0.1 N) at a sampling frequency of 100 Hz. In this system, the transducers transmitted pressure every 10 ms to the computer so that the participants’ center of gravity could be calculated, and the dynamic balance during a specific period could be obtained [19].

In this study, the following protocols were used: sit and stand (Sit-to-Stand), walk and return (Step/Quick Turn), and go up and down steps (Step Up/Over). Before starting the evaluation, personal data of the evaluated participant was filled with full name, age, gender, and body mass. The tests were repeated three times with an interval of 30 s between them.

Sit-to-Stand: Each participant was instructed to sit on a backless bench, feet apart, knees flexed at 90 degrees, and to stand up quickly and safely, remaining standing for a few seconds.

Step/Quick Turn: Each participant was previously instructed to walk on the platform, turn 180 degrees, and return to the starting point. The test started from the left side and then the right side.

Step Up/Over: To perform the test, a 20 cm step was placed in front of each participant, which they could see in front of them. The participants were instructed by the evaluator to step onto the step with their left leg and with their body in an upright position, stepping over it with their right leg, and stepping down with their left leg on the platform (floor) [19].

Randomization, allocation, blinding, and loss to follow-up. Group allocation was made, according to a randomized and stratified numerical sequence, generated before the first allocation to the groups. The study consisted of two parallel arms with a CG and a PG. Both groups did the same physical exercises. However, one group was supplemented with WP and the other with Control (Maltodextrin). An independent researcher, who did not know the numerical codes, prepared a random numerical sequence. The numerical sequence was kept in opaque envelopes and numbered sequentially, following the order generated by the software. The allocation to the groups was made by another independent researcher, who also did not know the numerical code that identified the groups.

Triple blind: Equally, the researcher who carried out the assessments, the one who carried out the interventions and the participants were blinded to the applied intervention.

The tabulation of data from all evaluation forms and the mathematical and statistical data treatment were also carried out by an independent researcher and blinded to the groups.

Losses to follow-up were treated by intention-to-treat, which was considered the appropriate analysis for testing “superiority” in randomized clinical trials [20]. In this approach, subjects were analyzed, according to their original allocation group, regardless of the treatment received, avoiding confounding bias caused by excluding non-adherent patients (pre-protocol analysis). Two participants dropped out of the study, one from each group; they declined due to personal issues (Figure 1).

Exercise training and supplementation: Nutritionists administered supplementation after training in both groups. Only they knew the allocation of participants, but they were not present during the RT session. Both PG and CG were supplemented immediately after each session, by ingesting 20 g of the respective supplement diluted in cold water.

Nutritional Guidance: All participants received nutritional guidance to standardize daily WP intake over the experimental protocol period. As recommended by the PROT-AG Study Group for older adults individuals (>65 years) [21], WP intake was from 1.0 to 1.2 g of protein/kg of body weight per day in the diet [22,23,24]. 

Resistance training (RT): The training sessions had a frequency of twice a week for 12 weeks. The RT protocol targeted the major muscle groups: pectoral press, leg press, leg extension, leg curl, plantar flexion, and abdominal crunch. In each exercise, three sets of 8 to 12 repetitions were performed. The intensity was first determined in 70% of the values obtained in the one-repetition maximum (1RM) test [25], and then adjusted during each session to 7–8, according to the OMNI Resistance Exercise Scale, which has values from 0 to 10, and has been proven to be able to track strength in older adults [26]. The intervals between sets were from one to two minutes. All sessions had a total time of 45 to 60 min, and there was direct supervision from a trained and experienced professional in all 24 sessions.

Statistical analysis: Data were stored and analyzed using the Statistical Package for the Social Sciences (SPSS) 24.0 program. The normality and homogeneity of the variances and the adherence to the Gauss curve were obtained, using the Shapiro-Wilk and Levene tests, respectively. The descriptive analysis of the sample was done, using the mean and standard deviation. The Student’s T-test was used to compare the quantitative sample’s baseline, and categorical variables were presented in terms of their absolute and relative values, and the chi-square test and/or Fisher’s exact test were used to verify the association. Parametric tests (two-way ANOVA) were used for the variables that satisfied assumptions (muscle strength, functional tasks, glycemic control, body composition, and handgrip strength). The accepted significance level was 5% (*p* ≤ 0.05). 

The calculation of the effect size was performed by calculating the square partial eta (ƞ2). The results were analyzed, according to the criteria: small (0.02), medium (0.13), and large (0.26).

The size of the sample was calculated through a pilot study with five T2DM older adults. The chosen variable was the primary outcome of muscle strength analyzed by the isokinetic peak torque (PT) assessment of the dominant limb quadriceps, with which we assumed the following two-tailed hypothesis: alpha value (error probability 1) of 5%; beta value, and test power of 80%. For these values, 13 patients were considered in each group.

## 3. Results 

Characteristics of the participants included in the study were not different between groups at the baseline (Table 1).

Regarding the muscle strength measured evolution of the loads in each proposed exercise: In the leg press exercise, there was a significant difference in time (F_3.48_ = 16.604; *p* < 0.001; η2 = 0.342), there was not in the group (F_1.48_ = 0.284; *p* < 0.596; η2 = 0.003), nor the group*time interaction (F_3.48_ = 0.268; *p* < 0.848; η2 = 0.008). In the calf exercise, there was a significant difference in time (F_3.55_ = 11.209; *p* < 0.001; η2 = 0.259); there was none for the group (F_1.55_ = 0.009; *p* < 0.924; η2 = 0.000), nor in the group*time interaction (F_3.55_ = 0.406; *p* < 0.749; η2 = 0.013). In the chest press exercise, there was a significant difference in time (F_3.24_ = 34.646; *p* < 0.001; η2 = 0.520), there was no difference in the group (F_1.24_ = 3.356; *p* < 0.070; η2 = 0.034), nor the group*time interaction (F_3.24_ = 1.217; *p* < 0.308; η2 = 0.003). In the leg curl exercise, there was a significant difference in time (F_3.8_ = 50.606; *p* < 0.001; η2 = 0.617); there was not for the group (F_1.8_ = 1.6866; *p* < 0.197; η2 = 0.017), nor in the group*time interaction (F_3.8_ = 0.007; *p* < 0.999; η2 = 0.000). In leg extensor, there was a significant difference in time (F_3.12_ = 19.449; *p* < 0.001; η2 = 0.378) and group (F_1.12_ = 8.994; *p* = 0.003; η2 = 0.086) and no group*time interaction (F_3.12_ = 0.082; *p* < 0.970; η2 = 0.003). For the abdominal exercise, there was a significant difference in time (F_3.23_ =24.539; *p* < 0.001; η2 = 0.434); there was none for the group (F_1.232_ = 0.559; *p* = 0.456; η2 = 0.006), nor in the group*time interaction (F_3.23_ = 0.188; *p* = 0.905; η2 = 0.006). In the chest press exercise, there was a significant difference in time (F_3.24_ = 34.646; *p* < 0.001; η2 = 0.520); there was none for the group (F_1.24_ = 3.356; *p* < 0.070; η2 = 0.034), nor the group*time interaction (F_3.24_ = 1.217; *p* < 0.308; η2 = 0.037) (Figure 2).

Regarding the functional tests, there were no significant differences in any variables of the Step Up/Over, Sit-to-Stand, and Step/Quick Turn tests (Table 2).

There was no significant difference between glycemic parameters, body composition, estimated glomerular filtration rate (MDRD), and HGS between groups (Table 3).

## 4. Discussion

The main findings of the present study show that RT improves muscle strength in T2DM, regardless of WP supplementation. However, the findings also suggest that RT does not improve the glycemic profile, especially when measured by HbA1c. Body composition or functional tasks also had no significant differences. However, we have demonstrated that WP supplementation during RT can be a safe intervention regarding renal function.

Our hypothesis was denied, as it was expected that the group that was submitted to WP supplementation would be superior to the CG, concerning muscle strength, functional tasks, glycemic control, and body composition. This advantage has been demonstrated in other studies with non-diabetic populations [14,27,28]. However, there was no significant difference between the groups in our sample of older adults living with T2DM.

These findings are analogous to the study of Miller et al. [29], which also tested RT with one hundred ninety-eight adults of both sexes, who were obese and living with T2DM. The intervention used WP supplementation, 20 g in the morning and 20 g after RT, in addition to 2000 IU of vitamin D daily, and it did not find differences between the groups. It is important to say that, in contrast to the present study, the control group did not receive isocaloric supplementation. According to the authors, the dose of 20 g of WP could have been insufficient to improve RT gains. 

In another study, Yang et al. [27] evaluated the acute effect of a unilateral strength exercise of the lower limb on the dose response to protein consumption, among thirty-seven non-diabetic older men, who consumed 0, 10, 20, and 40 g of WP. The participants were tested by biopsy. Even as it was possible to detect a small increase in protein synthesis in the lowest dose groups, 40 g of WP supplementation was superior in protein synthesis induced by RT when compared to the other groups. In young adults, they affirm that doses greater than 20 g do not lead to an improvement in the synergistic effect of supplementation on RT, which is different for this group of older adults. Therefore, it is possible to consider that the 20 g dose in the present study may have been insufficient to increase the results in morphology and function in the muscle tissue. Considering the aging process, which can also lead to less sensitivity to WP supplementation, the participants had a greater catabolic and lower anabolic propensity, caused by the pathophysiology of T2DM. It is equally important to consider that this group’s vulnerability is also due to the average number of drugs taken, and commonly having more than one associated chronic disease, which can affect the response of RT and/or WP supplementation. 

Our study also corroborates with Gaffney et al. [30], who investigated twenty-four men living with T2DM aged 55.6 (+/−5.7) years in a double-blind controlled clinical trial, where RT was combined with aerobic training, both intense and intermittent, three times a week for 10 weeks. WP supplementation was combined with carbohydrates and fat in a drink of 20 g, 10 g, and 3 g, respectively, and the control group consumed an isocaloric drink with 30 g of carbohydrates and 3 g of fat. Although the PG showed improvement in endothelial function and better blood perfusion, which in the long term may represent less muscular and neural damage, it did not show any difference in mitochondrial function or metabolic profile. It may be that the lack of gains in muscle strength, with or without WP supplementation, is related to chronic hyperglycemia and insulin resistance. Both are characteristics of T2DM and bring pathophysiological implications in the skeletal muscle itself, even if there are stimuli capable of promoting metabolic and morphological changes in healthy individuals.

The RT program did not improve functional tasks, such as going up and down stairs, walking and returning, and sitting and standing in both groups. This may also be related to the harmful effects of hyperglycemia on peripheral nerves, which causes demyelination and atrophy of motor axons, damage in mechanisms of neuron regeneration, leading to a reduction in the number of nerve fibers over the years, and also affecting the capacity of transmission of nerve action potentials [31]. Based on this, we can say that, neurologically, people living with T2DM are less trainable than healthy individuals of the same age. 

Ernandes et al. [32] also demonstrate that T2DM older adults, with or without peripheral neuropathy, have postural control deficits, especially in situations that prioritize the motor and visual systems. Consequently, functional tasks are potentially compromised, and this population is more susceptible to falls. These findings justify the recommendation for superior specificity in the elaboration of physical training or rehabilitation programs.

According to Mavros et al. [33], modifications in body composition, such as an increase in muscle mass and a decrease in fat mass, are necessary to promote changes in glycemia. In contrast, Cox et al. [34] defend that using a behavioral treatment with continuous glucose monitoring, aiming to minimize postprandial glucose excursion, can efficiently promote better glucose control, prior to any changes in body composition.

In our study, both groups experienced reductions in fat mass; however, these benefits were modest and unlikely to be clinically relevant. In addition, there were no thorough controls of postprandial glycemia or amplified lifestyle interventions, aiming to achieve better glycemic control. Moreover, in the present study, the values of HbA1c were 7.2 (+/−1.1) for the PG and 7.0 (+/−1.2) for the CG, which were within the recommended range for people living with T2DM, according to the American College of Physicians [35]. These results are consistent with other evidence that treatments targeting lower values of HbA1c require more rigorous interventions.

Although both groups were using drugs with protective and deleterious effects on muscle tissue and RT adaptations, the CG had more individuals using metformin, DPP4-inhibitors, as well as angiotensin-converting enzyme inhibitors (ACEIs) and angiotensin II receptor blockers (ARBs) that are recognized protective agents of the skeletal muscle [35,36]. Statins, more common in the PG, have the opposite effect, which may cause damage to the muscle and harm RT adaptations [34,35,36,37,38,39,40,41,42]. The magnitude of these influences and how much they were able to change the results, producing smaller adaptations in the PG, suppressing the potential effects of supplementation, and offering an advantage to the CG taking them to the same level of results, is difficult to determine.

According to Caplan et al. [43], 37% of people with diabetes also have renal disease, especially with increased amounts of time since diagnosis, and the age of the individuals. Therefore, concern about possible kidney damage due to higher WP intake is present among the professionals who work with this population. However, in the present study WP supplementation during RT was demonstrated to be a safe intervention regarding renal function.

The clinical implications of the present study showed the safety of the adopted protocol with nutritional guidance greater than 1g per kilogram of body weight per day (g∙kg^−1^∙d^−1^) and the use of supplementary protein in a group that did not have diagnosed renal function impairment. In addition, as there are few studies with older men living with T2DM in dietary approaches associated with physical exercises, the present study contributes to a greater understanding of the safety and efficacy of these types of interventions, given the specificities of the studied group.

The study’s limitations are related to the heterogeneity of the aging process and T2DM pathophysiology. This is a disease with multiple characteristics that implicate directly or indirectly the outcomes of nutrition or exercise interventions. We can emphasize the time elapsed since diagnosis, body composition, the severity of insulin insufficiency and/or insulin resistance, and the vast number of commonly associated comorbidities. These factors are distributed in such a diverse way, that it makes the stratification of a sample extremely difficult when we intend to create better levels of homogeneity and maintain generalizability.

The present study opens perspectives for further research on T2DM involving adequate amounts and variety of protein ingestion or WP supplementation, either by acute or chronic effect, and the best type, volume, and intensity of physical exercises, since the literature related to the subject is still insufficient.

## 5. Conclusions

In summary, both groups similarly increased their muscular strength. Supplementation with 20 g of WP after RT sessions did not show a clinically meaningful improvement on glycemic control, body composition, muscle strength, and functional tasks in the tested group. The protocol proved to be a safe intervention regarding renal function.

## Figures and Tables

**Figure 1 ijerph-20-05891-f001:**
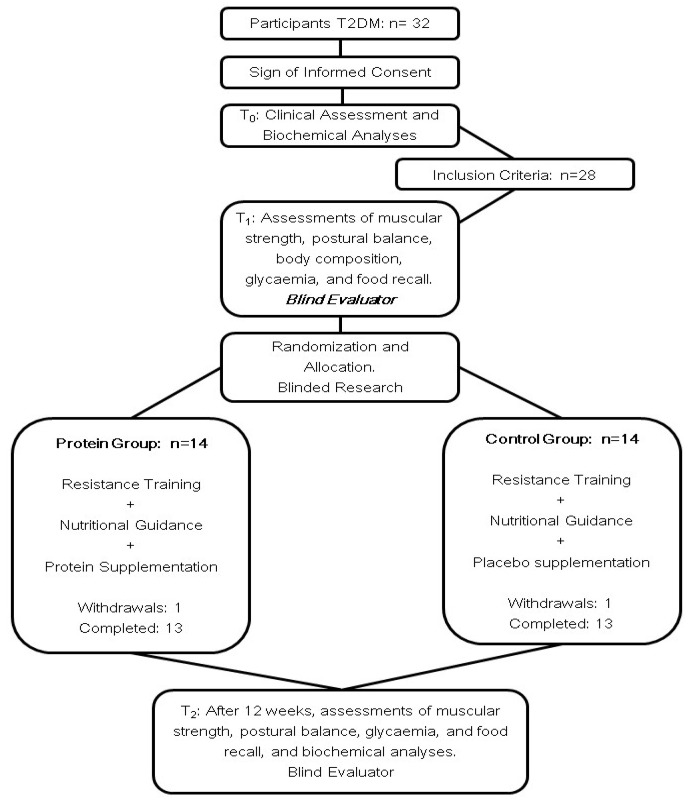
Flowchart of the study. Legend: T2DM—Type 2 diabetes mellitus; T_0_—pre evaluation; T_1_—initial evaluation; T_2_—final evaluation.

**Figure 2 ijerph-20-05891-f002:**
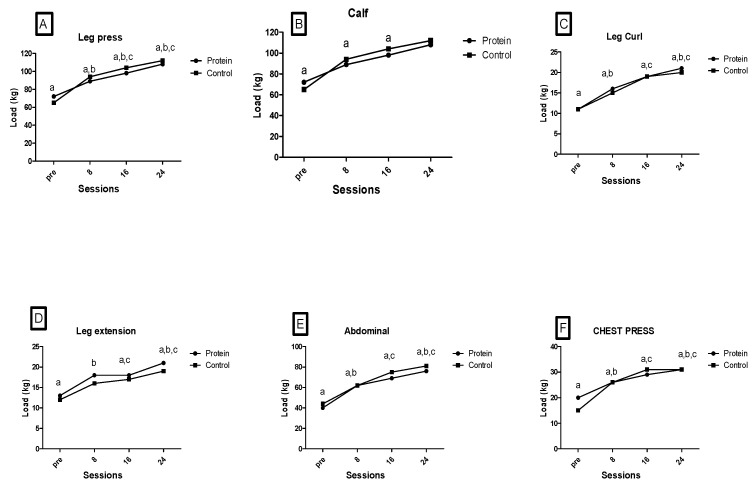
(**A**). Leg Press-Time-a ≠ pre and 8 (*p* = 0.001); pre and 16 (*p* = 0.001); pre and 24 (*p* < 0.001); b ≠ 8 and 24 (*p* = 0.016); c ≠ 16 and 24 (*p* = 0.016)]; (**B**). Calf-Time-a ≠ pre and 8 (*p* = 0.004); pre and 16 (*p* = 0.004); pre and 24 (*p* < 0.001); (**C**). Leg Curl-Time-a ≠ pre and 8 (*p* < 0.001); pre and 16 (*p* < 0.001); pre and 24 (*p* < 0.001); b ≠ 8 and 24 (*p* < 0.001); c ≠ 16 and 24 (*p* < 0.001); (**D**). Leg Extension-Time-a ≠ pre and 8 (*p* < 0.001); pre and 16 (*p* < 0.001); pre and 24 (*p* < 0.001); b ≠ 8 and 24 (*p* = 0.005); c ≠ 16 and 24 (*p* = 0.005); (**E**). Abdominal-Time-a ≠pre and 8 (*p* < 0.001); pre and 16 (*p* < 0.001); pre and 24 (*p* < 0.001); b ≠ 8 and 24 (*p* = 0.001); c ≠ 16 and 24 (*p* = 0.001) (**F**). Chest Press-Time-a ≠ pre and 8 (*p* < 0.001); pre and 16 (*p* < 0.001); pre and 24 (*p* < 0.001); b ≠ 8 and 24 (*p* = 0.001); c ≠ 16 and 24 (*p* = 0.001)]. Two-way ANOVA, Post Hoc Bonferroni *p* < 0.05.

**Table 1 ijerph-20-05891-t001:** Participants’ characteristics at the baseline.

	Protein M(SD)	Control M(SD)	*p*-Value
^#^ Age (years)	68.1 (4.5)	68.9 (4.1)	0.63
^#^ BMI (kg/m^2^)	29.3 (2.6)	26.8 (3.8)	0.07
^#^ Education (years)	11.5 (3.1)	11.5 (3.1)	0.39
^#^ Time since diagnosis (years)	12.7 (3.8)	12.8 (6.4)	0.86
^#^ Other diseases (n)	1.6 (1.4)	1.2 (0.9)	0.36
^#^ Medication (n)	3.8 (2.0)	3.4 (1.8)	0.62
^§^ Ethnicity	n (%)	n (%)	χ^2^ (*p*)
Caucasian	9/64.3	11/78.6	1.343(0.51)
Asian	1/7.7	0/0	
Black and Brown Skin	4/28.6	3/21.4	
^§^ BMI (kg/m^2^)	n (%)	n (%)	χ^2^ (*p*)
Low weight	1/7.1	1/7.1	0.650 (0.72)
Eutrophic	6/42.9	4/28.6	
Overweight	7/50	9/64.3	
* Oral Hypoglycemic Medication	n (%)	n (%)	Fischer’s exact (*p*)
Thiazolidinediones	2/14.3	0/0	0.48
Sulfonylureas	8/57.1	8/57.1	1.00
Biguanide	8/57.1	9/64.3	1.00
DPP-4 inhibitors	2/14.3	3/21.4	1.00
SGLT2 inhibitors	2/14.3	2/14.3	1.00
* Cardiovascular Medication	n (%)	n (%)	Fischer’s exact (*p*)
ARBs	1/7.1	4/28.6	0.32
ACE-Is	2/14.3	1/7.1	1.00
Statins	5/35.7	3/21.4	0.67

^#^ Student’s T-test was used to compare the quantitative variables. ^§^ Chi-square test and/or * Fisher’s exact test was used to compare the categorical variables. Note: Some research participants ingested more than one oral hypoglycemic agent and cardiovascular medication. Legend: BMI: body mass index, Underweight = up to 22, Eutrophic = from 22 to 27, Overweight = 27 or more. DPP-4 inhibitors—Dipeptidyl peptidase 4 inhibitors, SGLT2 inhibitors—Sodium-glucose cotransporter inhibitors two; ARBs—Angiotensin II Receptor Blockers, ACE-I—Angiotensin Converting Enzyme Inhibitors.

**Table 2 ijerph-20-05891-t002:** Intra and intergroup comparison of the functional tasks of the Protein vs. Control groups.

Parameters	Pre (sd)	Post (sd)	η2	ANOVA					
				Time Effect		Group Effect		Group × Time Effect	
				F	*p*	F	*p*	F	*p*
Lift Up left side									
Protein	36.64 (9.90)	36.82 (12.31)	0.01	0.44	0.50	0.02	0.86	0.52	0.47
Control	35.1 (8.95)	39.3 (10.68)							
Lift Up right side									
Protein	36.92 (7.23)	38.00 (9.27)	0.01	1.74	0.19	0.07	0.78	0.83	0.36
Control	33.8 (9.6)	39.6 (11.3)							
Moment Time left									
Protein	1.51(0.19)	1.69 (0.33)	0.01	0.74	0.39	3.15	0.08	0.91	0.34
Control	1.78(0.35)	1.77 (0.42)							
Moment Time right									
Protein	1.68 (0.30)	1.76 (0.56)	0.01	0.03	0.85	0.004	0.95	0.84	0.36
Control	1.80 (0.42)	1.67 (0.25)							
Impact Index left									
Protein	44.54 (13.58)	38.10 (11.53)	0.03	0.19	0.65	0.06	0.80	1.86	0.17
Control	40.57 (12.21)	43.85 (13.0)							
Impact Index right									
Protein	40.00 (14.33)	34.18 (12.66)	0.01	0.47	0.49	5.74	0.02	0.46	0.50
Control	47.26 (17.13)	47.23 (14.94)							
Weight transfer									
Protein	0.50 (0.19)	0.52 (0.26)	0.001	0.004	0.95	0.03	0.86	0.05	0.82
Control	0.51 (0.32)	0.48 (0.32)							
Sway Velocity									
Protein	3.50 (1.16)	3.71 (0.88)	0.008	0.004	0.95	2.47	0.12	0.36	0.55
Control	4.08 (1.45)	4.33 (1.72)							
Turn Time-SND									
Protein	2.20 (0.73)	2.12 (0.58)	0.007	0.95	0.33	1.29	0.26	0.32	0.57
Control	2.54 (0.87)	2.23 (0.60)							
Turn Time-SD									
Protein	2.31 (0.74)	1.85 (0.36)	0.02	0.003	0.95	4.31	0.04	1.40	0.24
Control	2.34 (0.85)	2.61 (1.10)							
Turn Sway-SND									
Protein	50.26 (13.10)	45.21 (10.9)	0.02	0.08	0.77	0.51	0.47	1.31	0.25
Control	43.5 (13.1)	46.7 (12.9)							
Turn Sway-SD									
Protein	50.26 (13.10)	45.21 (12.62)	0.02	0.08	0.77	0.51	0.47	1.31	0.25
Control	43.71 (12.64)	46.73 (12.72)							

Two-way ANOVA, Post Hoc Bonferroni *p* < 0.05. Legend: DS-dominant side; NDS-non-dominant side; η2: Eta squared; F-value of the F-test statistic (mean square of variables divided by the mean square of each parameter).

**Table 3 ijerph-20-05891-t003:** Intra and intergroup comparison of estimated glomerular filtration rate, glycemic control and body composition, and HGS between protein vs. control groups.

	Pre (sd)	Post (sd)	η2	ANOVA					
				Time Effect		Group Effect		Group × Time Effect	
				F	*p*	F	*p*	F	*p*
MDRD (mL/min/1.73 m^2^)									
Protein	86.7 (32.4)	90.1 (24.9)	0.04	0.00	0.98	1.87	0.10	0.13	0.71
Control	82.3 (17.6)	79.2 (11.4)							
FRUCTOSAMINE (µmol/L)									
Protein	315.2 (75.0)	285.0 (40.1)	0.02	2.00	0.16	1.55	0.22	0.59	0.81
Control	281.6 (50.6)	260.6 (30.8)							
GLUCOSE (mg/dL)									
Protein	140.1 (65.9)	124.8 (37.7)	0.00	0.87	0.35	0.08	0.93	0.00	0.98
Control	141.2 (47.3)	126.5 (17.8)							
INSULIN (µU/mL)									
Protein	11.0 (3.9)	12.4 (9.0)	0.02	0.16	0.68	1.82	0.18	0.73	0.39
Control	17.6 (9.1)	14.3 (5.4)							
HbA1c (%)									
Protein	7.2 (1.1)	6.9 (1.0)	0.01	1.24	0.27	0.09	0.75	0.02	0.87
Control	7.0 (1.2)	6.5 (0.6)							
HOMA IR									
Protein	4.1 (1.7)	4.2 (2.6)	0.03	1.50	0.21	2.50	0.11	1.7	0.20
Control	6.5 (3.9)	4.1 (2.1)							
Body Composition									
LEAN MASS (kg)									
Protein	32.8 (5.3)	32.5 (4.8)	0.04	0.17	0.67	0.84	0.36	0.20	0.65
Control	34.3 (5.4)	34.4 (5.7)							
FAT MASS (kg)									
Protein	22.5 (7.8)	21.9 (8.1)	0.00	0.02	0.86	3.00	0.08	0.00	0.93
Control	26.8 (10.3)	26.6 (10.2)							
Hand Grip Strength									
HGS–DS (kg/f)									
Protein	36.7 (9.9)	36.1 (8.7)	0.01	0.34	0.55	0.20	0.65	068	0.41
Control	33.8 (8.7)	37.0 (4.6)							
HGS–NDS (kg/f)									
Protein	33.7 (10.8)	35.8 (10.3)	0.001	1.20	0.27	0.25	0.61	0.05	0.82
Control	32.0 (7.4)	35.1 (4.3)							

Two-way ANOVA, Post Hoc Bonferroni *p* < 0.05. Legend: MDRD-Modification of Diet in Renal Disease; HbA1c-Glycated hemoglobin; HGS-handgrip strength; LN-dominant side; NDS-non-dominant side; η2: partial Eta squared; F-value of the F-test statistical test (mean square of variables divided by the mean square of each parameter).

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
