# Peer review of "The Influence of Whey Protein on Muscle Strength, Glycemic Control and Functional Tasks in Older Adults with Type 2 Diabetes Mellitus in a Resistance Exercise Program: Randomized and Triple Blind Clinical Trial"

_ijerph, 2023, doi:10.3390/ijerph20105891_

Round 1
Reviewer 1 Report
Dear authors,
congratulations on your work. You tried to see if the supplementation of WP would result in higher increases. In general your work needs substanticial improvemment in the discussion since you compare your most important results with only three studies.
Here are other suggestions:
1.Your introduction did not describe the importance of your study.
2.In methods you have to describe inclusion and exclusion criteria. Its also not clear your methods, its confusing to understand what have you done.
3.In the statistical analysis you need to perform sample size calculation
4. You also need to improve the language in your manuscript and format according to IJERPH.
Author Response
Reviewer: 1
Thank you for all your comments and suggestions. All queries answers are marked in the text with yellow highlight color. We remain available for any further questions.
Reviewer #1:
Query 1: Your introduction did not describe the importance of your study
Response: We include /rephrased the following paragraphs :
Recently a systematic review comprised of 11 studies was published combining diet, supplements, and exercises [16], and in only two of them, protein supplementation [17,18] in people living with DMT2 did not respond to a certain type of nutritional stimulation. However, on a positive note, recent data have shown that increasing the dose of the supplement or extending the intervention period and type of exercise can lead to better clinical outcomes, which should be investigated further, therefore, more studies are needed to better understand how to optimally combine these three factors for the management of T2DM.
The positive effects of RT, WP supplementation, and the combination of both in glycemic control, body composition, metabolic profile, inflammation, strength, and potential balance are well documented [16–20]. Accordingly, the importance of this study resides in the contribution that it can offer for a better understanding of doses, frequency, volume, and intensity of these resources to provide more efficient and safe strategies that can benefit the globally increasing population of older adults living with T2DM.
Query 2: In methods you have to describe inclusion and exclusion criteria. Its also not clear your methods, its confusing to understand what have you done.
Response: We detailed more the inclusion and exclusion criteria
Query 3: In the statistical analysis you need to perform sample size calculation
Response: We include sample size calculation.
Query 4: You also need to improve the language in your manuscript and format according to IJERPH.
Response: The text was review for author native an certified American Translators Association professional.
Reviewer 2 Report
Dear editor in chief
IJERPH
After to review the manuscript entitled “
The Influence of Protein Supplementation on Muscle Strength, 2 Glycemic Control and Functional Tasks in Older Adults with 3 Type 2 Diabetes Mellitus in a Resistance Exercise Program: 4 Randomized and Triple Blind Clinical Trial”. The objective of this study was to determine de influence of protein supplementation on efforts made by elderly people with diabetes
Theme of this manuscript is interesting and well writing, but there are some question and suggestions that I want to express.
Some specific highlights, review some spelling words.
The title of this manuscript indicates protein supplementation, but in methodology did not describe the source of protein.
Line 86, it’s possible to age average or range should be put a (-) if use range or (±) if prefer to utilize average and standard deviation.
Line 167. Capital letter the first word. Review title Figure 1. Its possible to add what meaning the acronyms used in this figure.
Line 230. In Figure 2, add x-axis name.
Line 241. Review all tables for express significance p or P.
Lines 186 to 197. Maybe it is necessary to review statistical analysis and describe what variables were analyzed by what method, and put the statistical model used to data analysis. Maybe it is possible to change the analysis for repeated measured trough the time for the effort analysis, or use a mixed model. Then if they change the statistical analysis the authors need to change results and discussion.
Author Response
Reviewer: 2
Thank you for all your comments and suggestions. All queries answers are marked in the text with yellow highlight color. We remain available for any further questions.
Reviewer #2:
After to review the manuscript entitled “
The Influence of Protein Supplementation on Muscle Strength, 2 Glycemic Control and Functional Tasks in Older Adults with 3 Type 2 Diabetes Mellitus in a Resistance Exercise Program: 4 Randomized and Triple Blind Clinical Trial”. The objective of this study was to determine de influence of protein supplementation on efforts made by elderly people with diabetes
Theme of this manuscript is interesting and well writing, but there are some question and suggestions that I want to express.
Some specific highlights, review some spelling words.
Query 1: The title of this manuscript indicates protein supplementation, but in methodology did not describe the source of protein.
Response: We include in the title: ‘’Whey protein’’ and change it in the text.
Query 2: Line 86, it’s possible to age average or range should be put a (-) if use range or (±) if prefer to utilize average and standard deviation.
Response: We include in the text: with aged ±68.2(4.0) between 65 and 78 years old.
Query 3: Line 167. Capital letter the first word. Review title Figure 1. Its possible to add what meaning the acronyms used in this figure.
Response: Figure 1. Flowchart of the study.
Legend: T2DM - Type 2 diabetes mellitus; T0 - pre evaluation; T1 - initial evaluation; T2 - final evaluation
Query 4: Line 230. In Figure 2, add x-axis name.
Response: Session
Query 5: Line 241. Review all tables for express significance p or P.
Response: We standardize p.
Query 6: Lines 186 to 197. Maybe it is necessary to review statistical analysis and describe what variables were analyzed by what method, and put the statistical model used to data analysis. Maybe it is possible to change the analysis for repeated measured trough the time for the effort analysis, or use a mixed model. Then if they change the statistical analysis the authors need to change results and discussion.
Response: We readjust the tex and ad more detailed justification of the statistical analysis: (Line 229) The descriptive analysis of the sample was done, using the mean and standard deviation. The Student T-test was used to compare the quantitative sample's baseline, and categor-ical variables were presented in terms of their absolute and relative values, and the chi-square test and/or Fisher's exact test were used to verify the association. Parametric tests (2-way ANOVA) were used for the variables that satisfied assumptions (muscle strength, functional tasks, glycemic control, body composition, and handgrip strength). The accepted significance level will be 5% (p ≤ 0.05).
(Line 239) The size of the sample was calculated through a pilot study with five T2DM older adults. The chosen variable was the primary outcome of muscle strength analyzed by the isokinetic peak torque (PT) assessment of the dominant limb quadriceps, which we assumed the following two-tailed hypothesis: alpha value (error probability 1) of 5%; beta value, and test power of 80%. For these values, 13 patients were considered in each group.
We include at the end of table 1:
Student T-test was used to compare the quantitative variables
- Chi-square test and/or *Fisher's exact test to compare the categorical variables
And end of table 2, 3 and figure 2 Two way, ANOVA, Post Hoc Bonferroni p<0.05